# Constructing and Testing AI International Legal Education Coupling-Enabling Model

**Yunyao Wang** [1,2] **and Shudong Yang** [1,*]

1   School of Law, Chongqing University, Chongqing 400044, China; wangyunyao@nsu.edu.cn
2   School of Information and Business Management, Chengdu Neusoft University, Chengdu 611844, China
*   Correspondence: ysd@cqu.edu.cn

**Abstract:** In this paper, we aim to assess the coupling capability of artificial intelligence in international legal education, delving into crucial aspects of its implementation and effectiveness. This paper constructs a coupling empowerment model of AI international legal education by using artificial intelligence technology. It also discusses the application of Pearson product–moment correlation coefficient in correlation analysis, the implementation of AI knowledge mapping in the help of intelligent parents, and the application of BP neural algorithm in artificial neural networks in order to establish a cognitive student model. This teaching mode can provide personalized learning experience and intelligent teaching support and allow accurate assessment of students' learning level and cognitive ability. The results show that the employment rate of students is increased from 75% to 100%, and the evaluation of practicability is maintained at 10 points. It proves that AI technology provides an innovative approach to international law education, which is expected to promote the efficient use of educational resources and improve students' performance and employment rate.

**Keywords:** artificial intelligence; international legal education; backpropagation neural algorithm; learning behavior analysis; coupling-enabling model

## 1. Introduction

At present, science and technology is the primary productive force, and artificial intelligence technology has been widely applied to all fields of human society production and life and is profoundly changing the pattern of economic and social development [1]. In order to effectively grasp this key historical opportunity, a lot of important techniques have been formed in the development of AI layout, and education has become one of the key links. In recent years, a series of national and local policies to promote the development of AI have emerged, covering the new requirements for the development of education [2]. As a code of conduct regulating social relations, law needs to face and respond positively to the influence of artificial intelligence on various industries.

International law education in universities, as a developing force of the legal profession, provides legal professionals for the legal profession and all fields of society [3]. Due to the traditional mode of operation of the legal profession, this has indirectly transformed into a rapid demand for interdisciplinary professional strength of international law. However, the traditional teaching method is mainly based on teachers' curriculum teaching, which emphasizes one-way inculcation [4]. Therefore, the cultivation of legal talents in the age of artificial intelligence needs to meet the requirements of intellectualization [5]; as such, we should promote the reform of the traditional international law education teaching mode and construct a law education teaching model matching the talent training system of artificial intelligence law [6]. How to adapt to the new era and new requirements, actively exploring and effectively responding to the impact and challenges brought by the development of artificial intelligence technology on legal education, is the proposition of the times that legal educators must face [7].

In this paper, an AI international legal education coupling empowerment model is constructed based on AI-related technology; firstly, Pearson correlation coefficient is applied to carry out correlation analysis between different learning behaviors of learners and measure the linear correlation between variables. Secondly, it realizes the function of intelligent parenting assistant through AI knowledge mapping, and then applies a BP neural algorithm in an artificial neural network, combined with cognitive theory, to establish a cognitive student model that can reflect the learning level and cognitive ability of students. Finally, in the practical analysis, the effectiveness of the AI-coupled empowerment model is verified through system testing, AI teaching coupling analysis, and long-term impact and sustainability analysis, which proves that the AI technology provides new kinetic energy for the transformation and upgrading of legal education, prompts the legal education system to realize all-around changes and innovations, and improves the quality of legal education.

## 2. Literature Review

Ouyang, F. et al. [8] proposed that AI has different roles in education: one is as a knowledge model and tutor to help learners better understand and absorb knowledge. The second is as a support tool for learning, collaborating with learners to accomplish learning tasks together. Thirdly, AI also empowers learners and allows students to become learning agents. Villegas-Ch, W. et al. [9] proposed the use of AI to analyze student-generated data, categorize patterns of student needs, and make decisions that benefit each student's learning. The use of structured knowledge and experiences that mimic human thought processes leads to better meeting the individual learning needs of students. Alam, A. [10], by assessing the impact of AI technologies on teaching and learning, concluded that AI produces positive effects in terms of both improving the quality of teaching for teachers and promoting learning outcomes for students. The possible challenges of AI applications in education are explored, while highlighting the great potential of AI in helping schools to improve the quality of teaching and learning, which can help to promote innovation in the field of education. In the work by Chan, C. K. Y. [11], for the purpose of formulating an education policy on AI in higher education, a quantitative and qualitative research methodology was used to survey 457 students from different disciplines in Hong Kong universities and 180 faculty members, presenting a comprehensive policy framework for AI eco-education that includes pedagogical, managerial, and operational aspects. The focus is on enhancing teaching and learning with the help of AI on issues such as handling and privacy, security and accountability, and infrastructure and training. Borenstein, J and Howard, A [12] suggest the need to rethink the content of future training for developers, designers, and professionals in AI, which is achieved through a more comprehensive and systematic incorporation of AI ethics into the curriculum. In the paper, different approaches to AI ethics are presented and a series of recommendations related to the teaching of AI ethics are made.

Perrotta, C et al. [13] examined the Khan Academy and ASSISTments Intelligent Tutoring System, illustrating instances of AI elements. The scholarly work of numerous data scientists utilizing deep learning to forecast facets of educational achievement was thoroughly explored, drawing on research in science and technology. Holmes, W et al. [14] conducted an investigation involving 60 leading researchers in the AI and educational development field, exploring ethical and application issues associated with AI in education. Recognizing the lack of training among most AI education researchers to address emerging ethical concerns, there is a particularly crucial need to effectively integrate multidisciplinary approaches with AI. Nemorin, S et al. [15] utilized text mining and thematic analysis to scrutinize key themes emerging in AI education in recent years. The findings are categorized into three segments: achieving geopolitical dominance through education and technological innovation, developing and expanding niche market strategies, and altering management narratives, perceptions, and norms. Knox, J [16] firstly analyzed two crucial policy documents issued by China's central government, emphasizing the pivotal

role educational institutions play in national and regional AI development strategies. Subsequently, three key private education companies instrumental in the advancement of educational AI applications in China are presented. Finally, it is demonstrated that while government policies allocate a significant role for education in the national AI strategy, the private sector is capitalizing on favorable political conditions to swiftly develop educational applications and markets.

## 3. Constructing AI International Legal Education Coupling-Enabling Model

### 3.1. Construction of AI Teaching Model

International legal education has always been accompanied by legal development and social development, and has carried out constant mutual combination in both legal technology and science and technology. The popularization of artificial intelligence has profoundly changed the form of knowledge expression, access methods, and dissemination pathways, providing new development momentum and new development prospects for legal education, but also bringing unprecedented challenges. The application of AI technology has, to a certain extent, pushed forward the integration process of the artificial intelligence model and the education system, making the mechanism of education also realize a comprehensive transformation. The cultivation of talents should not only focus on the traditional knowledge structure but also on the application of AI technology and technical processing methods to ensure that it can provide maximum convenience for educational work.

Figure 1 shows the structure of the international law intelligent education system, which firstly focuses on the richness and advancement of the course content in the construction. Advanced teaching cases are the basis for constructing the online teaching platform, which provides students with a more in-depth and practical learning experience. These cases are not only the inculcation of theoretical knowledge, but also the demonstration of the operation and problem-solving ability of jurisprudence in practical application through real cases. Secondly, the system makes full use of the mature network technology to build the platform framework to ensure that the course content can be efficiently and smoothly displayed on the network platform. This requires not only the skillful use of network technology, but also a deep understanding of the field of jurisprudence in order to present the knowledge to students in the most appropriate way. Through such a platform, not only is the sharing of course resources realized, but a wider range of academic resources are also applied to the students. An AI education system is characterized by efficiency, remoteness and resource sharing. Students are able to obtain the latest and most comprehensive knowledge of law remotely through this system, while realizing the sharing and application of course resources. This feature makes students no longer limited by geographical and time constraints in the academic field and allows them to arrange their study time and subject combinations more flexibly to achieve the goal of personalized learning.

This system is not only a traditional educational platform but also a platform for academic communication. Students can participate in the flipped classroom teaching mode and customize their independent and personalized learning plans in the online learning space. At the same time, students can also engage in interactive and shared communication through this platform to promote academic cooperation and exchange. This provides a new auxiliary teaching means for traditional teaching and promotes the innovation and development of the coupling mode of international legal education.

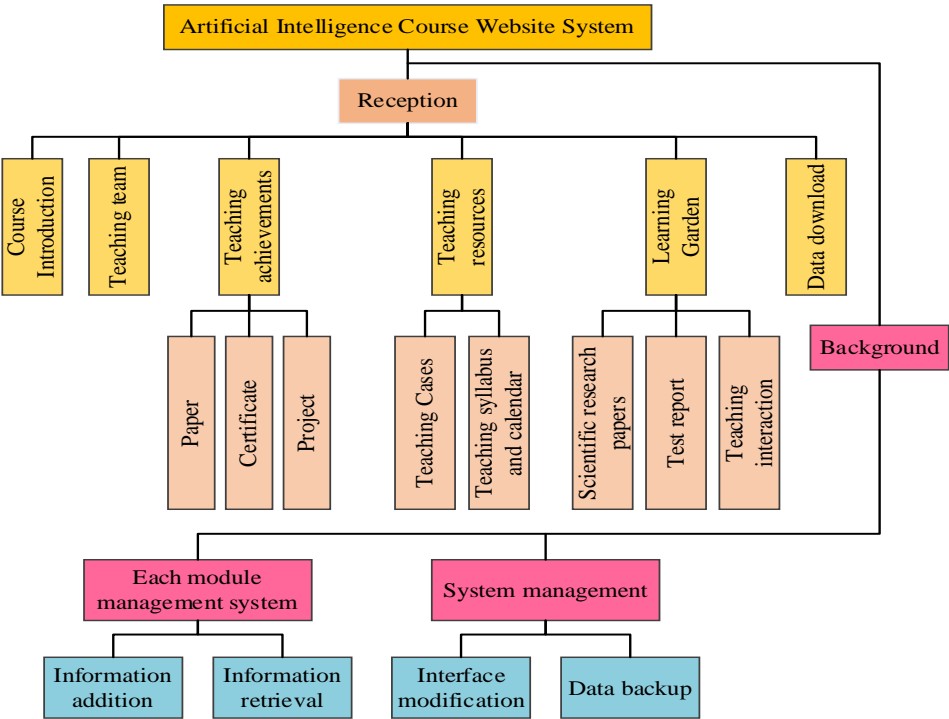

**Figure 1.** Structure of intelligent education system of international law.

*3.2. Decision Reasoning in Intelligent Teaching*

The teaching decision reasoning of the intelligent education system for international law is a critical task that not only relies on the various information collected by the system, but also requires the deep integration of AI technology to achieve a more personalized and efficient education model. In the innovative exploration of the coupled empowerment model of AI and international legal education, the teaching decision reasoning process becomes more intelligent and adaptable. The teaching decision reasoning process is shown in Figure 2, where the client's role is not only to provide an interface with the user, but also to analyze students' learning, academic level and personalized needs in real time through AI technology. Such an intelligent client can more accurately reflect students' learning characteristics and provide more detailed information for subsequent teaching decisions. At the server level, with the development of AI technology, the functional modules of teaching decision reasoning become more intelligent and adaptive. For example, the system can analyze students' past learning behaviors through machine learning algorithms and predict students' future academic development direction, thus providing teachers with more targeted teaching suggestions. At the same time, the server side can also realize intelligent management of educational resources based on big data technology to ensure that students can obtain learning content that best suits individual differences. The role of the database in this intelligent education system is also further strengthened. In addition to accomplishing data storage and management, the database is also able to support the system's continuous optimization and upgrading of the teaching decision-making model through the accumulation of a large amount of student learning data. This data-driven optimization process helps to ensure that the system is able to better meet the needs of international legal education through continuous innovation.

The innovative exploration of the coupling and empowerment mode of AI and international law education will make the international law intelligent education system better adapt to the personalized learning needs of students, enhance the level of education services, and realize the intelligence and optimization of teaching decision reasoning.

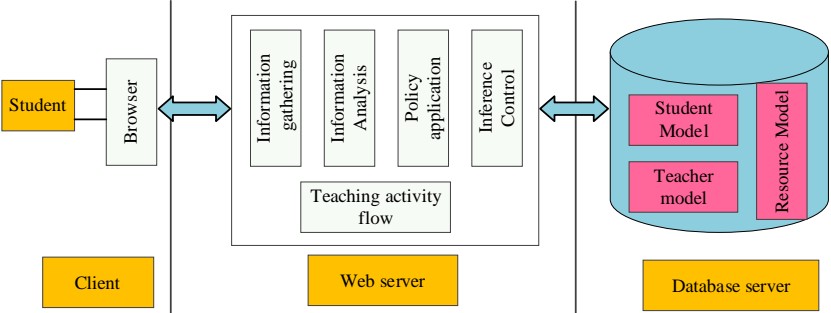

**Figure 2.** Teaching decision reasoning process.

## 4. AI Intelligent Technology

### 4.1. AI Data Analysis

With the change of the international law education mode supported by the intelligent teaching ecosystem based on AI technology, there is a move from the traditional classroom online to strengthen the classroom interaction, and then to intelligent teaching [17]. Through the collection of students' whole process behavioral data, with the help of big data statistical analysis methods [18], an analysis report of students' learning situation is formed, and AI data analysis technology is used to reasonably label students with intelligent classification methods. Labeling refers to grouping users according to their basic attributes, classroom performance, learning level, test level, etc., to accurately identify the degree of mastery of a particular student's knowledge points and to realize an accurate portrait of student learning. Then, the intelligent teaching ecosystem combines the knowledge reasoning function of the knowledge map to provide students with intelligent legal knowledge points, such as pushing, assigning personalized homework, etc., and provide intelligent teaching assistants to assist students in their learning.

Figure 3 shows the AI data learning behavior analysis mode, the model of which is based on the learning behavior data of international law students when using the AI teaching system, through different analysis processes, and finally the analysis results, and the results are fed back to the learning stakeholders. That is, the analysis model consists of learning data collection, data analysis [19], results presentation of the three basic links, through the storage of data, data collection, and processing, visual presentation of the results to achieve the process of learning behavior analysis, according to the results of the analysis of the results of the learner, and then provide intervention and feedback and a series of related learning guidance; learning behavior analysis is a cyclical process of continuous improvement.

The computational model established through Pearson is shown as follows:

$$r = \frac{\sum (X - \overline{X})(Y - \overline{Y})}{n \times \sigma_x \times \sigma_y}$$

In the formula, $\sigma_x$ and $\sigma_y$ represent the standard deviation of the two variables, $n$ is the capacity of the sample, and the meaning of $n$ is the sum of the product of the two standardized scores divided by the sample capacity. The value of $r$ is between $-1$ and $+1$: the positive value of $r$ indicates that the correlation between the two is positive, and the negative value of $r$ indicates that the correlation between the two is positive.

After analyzing the data of learners' learning behaviors, this is used to explore the relationship that exists between various learning behaviors of the learners. In this paper, Spearman correlation coefficient analysis is employed to examine the correlation between various learning behaviors exhibited by learners. This method assesses the linear relation-

ship between variables, and notably, its calculation model does not necessitate adherence to a normal distribution for the data. The formula for calculation is as follows:

$$r = 1 - \frac{6 \sum\limits_{i=1}^{n} d_i^2}{n^3 - n}$$

The application of these two correlation coefficients helps to reveal the relationship between various learning behaviors of learners in the innovative exploration of the coupled empowerment model of AI and international legal education, providing a powerful tool for further analysis.

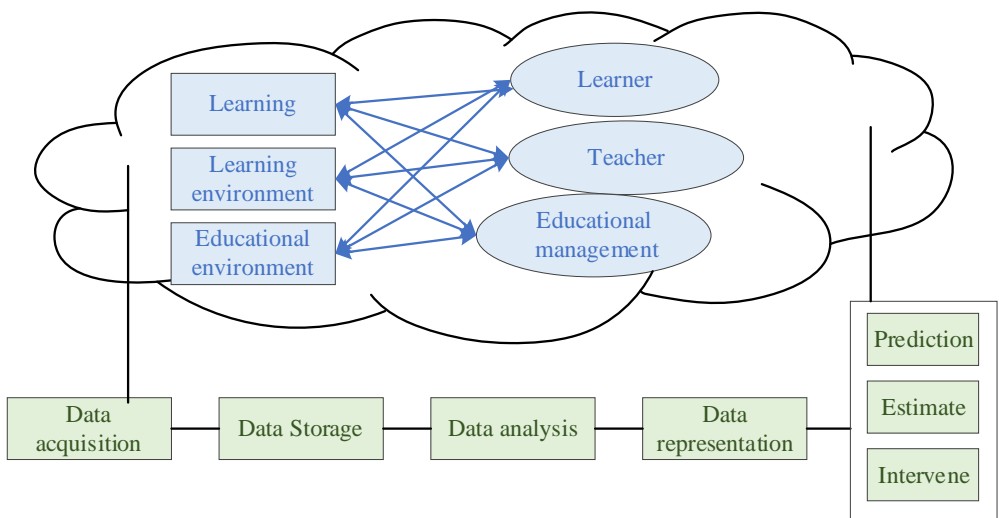

**Figure 3.** AI data learning behavior analysis model.

### 4.2. AI Knowledge Graph

Knowledge mapping mainly focuses on how to use AI-related technology to acquire structured domain knowledge to meet the systematic knowledge base organization and management needs of intelligent parenting assistants, and at the same time, on the basis of the knowledge provided by the knowledge mapping for the analysis and determination of parenting problems, it combines the theories of psychology, pedagogy, and sociology to construct a comprehensive solution model for parenting problems. Figure 4 shows the architecture of the AI intelligent knowledge mapping system, which is mainly divided into the data collection layer [20], the knowledge mapping layer, and the dialog system layer; the data collection layer is responsible for the acquisition and management of international law education problems and theoretical data [21]. The knowledge mapping layer is responsible for constructing the knowledge map based on the data collected in the data layer. The Dialogue System Layer is mainly based on the structured domain knowledge provided by the Knowledge Graph and uses AI technology to realize the function of intelligent education assistant. The knowledge graph mainly provides domain knowledge for the AI education system and supports the dialog system. The construction of the knowledge graph mainly includes graph schema definition, knowledge acquisition and knowledge fusion. Based on the information of the three factors of problem behavior, internal individual characteristics, and external environment, the AI technology summarizes the reasons for the emergence of the problem and gives the solution countermeasures, relevant theoretical knowledge, and related cases.

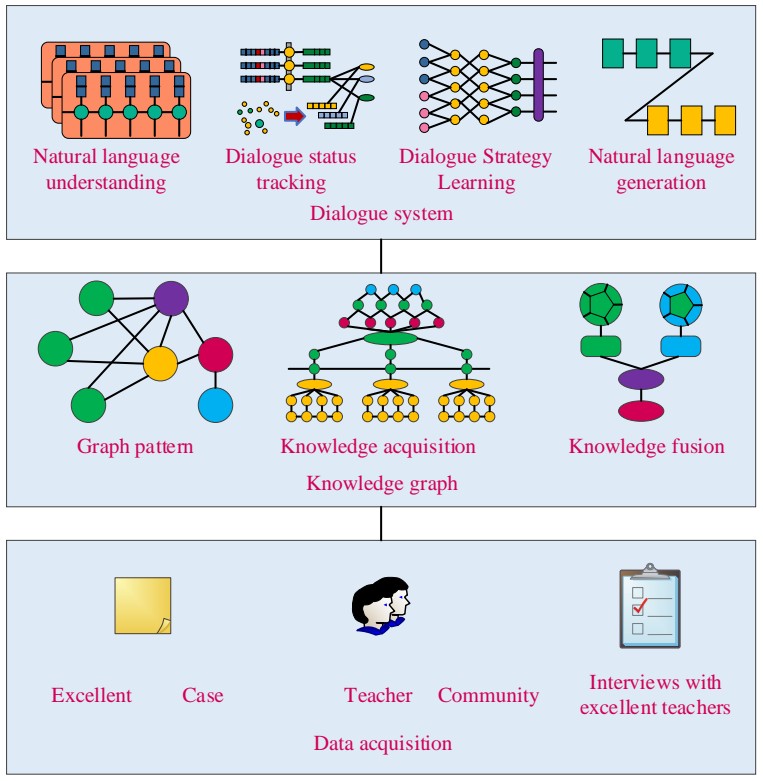

**Figure 4.** AI intelligent knowledge mapping system architecture.

*4.3. AI Intelligent Diagnosis*

AI intelligent teaching is a new teaching idea and teaching method, in which the simulation is the teacher and the service object is the student, that aims to synthesize the theory of educational psychology and cognitive science based on the characteristics of the learner and the state, tracking the changes in the characteristics of the learner and the state and automatically generating teaching information and adjusting the teaching process and teaching strategy [22]. Figure 5 shows the structure of AI intelligent diagnostic system in which the domain model stores the specialized knowledge of the course taught to the students, which can generate questions and provide correct answers to the questions and the process of solving the problems. The diagnostic model analyzes the student's response using diagnostic rules to determine what the student already knows or what misconceptions the student has generated and passes them on to the current state of the student model. The role of the teacher model is to incorporate knowledge of instructional strategies and lesson structure to select questions for the student to answer, to monitor and evaluate their behavior, and to select appropriate remedial materials for the student when needed. The cross-interpretation model in the teacher model, as well as the student model, is the main means of realizing that individuals teach in an interactive way [23].

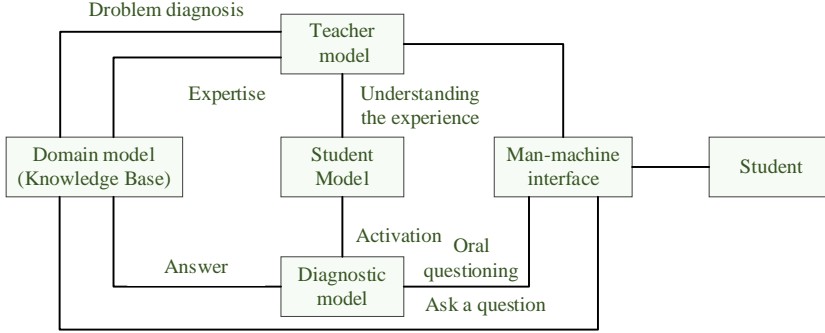

**Figure 5.** AI intelligent diagnosis system structure.

Using the BP neural algorithm in an artificial neural network, combined with the cognitive theory, a cognitive student model that can reflect the learning level and cognitive ability of students is established [24].

The basic structure of BP neural network is a feedforward neural network with more than three layers, mainly using the BP algorithm to solve the problem of hidden layer errors that cannot be calculated due to not being directly connected to the outside world. The BP algorithm belongs to supervised learning algorithms and is an effective method for calculating the derivative of large-scale systems composed of multiple basic subsystems [25]. The structure of the BP neural network is shown in Figure 6. The network trains $(X_k, Y_k)$ through samples to complete learning. If the *k*-th pair of samples is provided to the network, the output error will be $\delta_j^{(k)} = y_j^{(k)} - {}_j^{(k)}$. *j* is the *j*-th component of the actual output of input sample $X_k$, ${}_j^{(k)} = f\left(\sum_r W_{rj} \cdot b_{rk} - \theta_j\right)$, *r* is the number of hidden layer neurons, $\theta_j$ is the threshold of the *f*-th neuron in the output layer, $b_{rk}$ is the sigmoid function, *r* is the net output of hidden and neurons, and $b_{rk} = f\left(\sum_i W_{ir} \cdot a_i^{(k)} - T_r\right)$ [26]. The mean squared error of the output layer for sample *k* is $E_k = \frac{1}{2}\sum_j^n \left(y_j^{(k)} - {}_j^{(k)}\right)^2$, and $E_k$ is the number of output layer units, which decreases gradually with $E_k$ correction of connection weights [27].

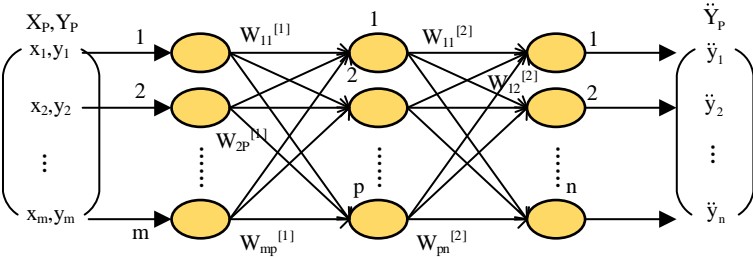

**Figure 6.** Bp neural network structure.

The input of the network includes 15 quantities as input nodes by selecting six levels of cognitive activities, i.e., the judgment values of six aspects of literacy, comprehension, application, analysis, synthesis and evaluation, and test scores, as well as age, education level [28], physiological conditions, learning environment, mood, learning efficiency, etc., and the output nodes by selecting the degree of mastery of students in terms of concepts, skills, and applications. It actually accomplishes a nonlinear mapping from a 15-dimensional space to a 3-dimensional space, i.e.:

$$X \to Y$$
$$f_{bp}\ X = \{X_1, X_2, X_3, \cdots, X_{15}\}$$
$$Y = \{Y_1, Y_2, Y_3\}$$

In the above equation, $X_1, X_2, \cdots, X_6$ represent the assessment values of six levels of cognitive activities, respectively, $X_7$ represents the test scores, and $X_8, X_9, \cdots, X_{15}$ represent age, education, physiological conditions, learning environment, mood, and learning efficiency, respectively. $Y_1, Y_2, Y_3$ denote students' mastery of concepts, skills, and applications, respectively [29].

The BP neural network consists of an input layer, an output layer, and a hidden layer. The input to the input layer is denoted as $x_i$ and its output is expressed as follows:

$$O_i^{(1)} = x(i),\ i = 1, 2, \cdots, n$$

with $w_{ij}^{(2)}$ and $f[\cdot]$ representing the weight coefficients of the implicit layer of the BP neural network and denoting the mapping function, the computation formulas for its input and output are respectively expressed as follows:

$$net_i^{(2)}(k) = \sum_{j=1}^{m} w_{ij}^{(2)} O_j^{(1)}(k)$$

$$O_i^{(2)}(k) = f\left[\text{net}_i^{(2)}(k)\right]$$

with $w_{li}^{(3)}$ and $g[\cdot]$ representing the weight coefficients of the output layer of the BP neural network and denoting the mapping function, the computation formulas for its input and output are respectively expressed as follows:

$$net_l^{(3)}(k) = \sum_{i=1}^{p} w_{li}^{(3)} O_i^{(2)}(k)$$

$$O_l^{(3)}(k) = g\left[net_l^{(3)}(k)\right]$$

for the $p$ st sample, whose actual and network outputs are $O_p(k+1)$ and $O'_p(k+1)$, respectively, then the error is given by the following:

$$E_p = \frac{1}{2}\left[O_p(k+1) - O'_p(k+1)\right]^2$$

Through the above steps, the BP neural network can be trained according to the input data and expected output of the students, and the weights can be continuously adjusted, so that a cognitive student model can be created that reflects the learning level and cognitive ability of the student [30]. This model can output the student's mastery of concepts, skills, and applications through the learner's cognitive activity assessment values and other relevant information, which supports the subsequent creation of a coupled empowerment model of AI and international legal education [31].

## 5. Practical Analysis of the AI Model of International Legal Education

### 5.1. System Testing

We examined current instances of AI application in legal education to gain insights into the current landscape and draw from successful experiences. To facilitate analysis, the 5000 data sets were subdivided into five datasets. Dataset 1 comprised 5 categories with 1000 pieces of data, dataset 2 comprised 10 categories with 2000 pieces of data, dataset 3 included 15 categories with 4000 pieces of data, and dataset 4 encompassed 20 categories with 5000 pieces of data. These datasets were subjected to classification using an intelligent educational model, the K-means teaching model, and a Support Vector Machine Teaching System. The experimental results from data mining are presented in Table 1.

**Table 1.** Results of experimental data mining.

| Data Set | K-Means Instructional Model | Teaching System of Vector Machine | AI Education Model |
|---|---|---|---|
| Data set 1 | 69.5% | 82.6% | 99.5% |
| Data set 2 | 62.4% | 78.4% | 98.8% |
| Data set 3 | 60.2% | 74.2% | 97.9% |
| Data set 4 | 54.7% | 68.2% | 96.4% |

Based on the experimental findings, the AI international law education model demonstrates an impressive classification accuracy of 99.5%. Even with a substantial volume of data and a diverse range of categories, the accuracy remains remarkably high at 96.4%. This surpasses the performance of both the support vector machine teaching system and the

K-means teaching model, showcasing the model's superior ability to precisely extract learning knowledge for users. Furthermore, the algorithm operates efficiently, completing tasks in a short timeframe, significantly outperforming manual processing times and alternative algorithms. With the integration of AI technology into the international law education system, the benefits extend beyond recommending learning programs. The system can analyze user learning behaviors to discern individual interests, subsequently providing tailored recommendations for relevant courses. The intelligent education system enhances the automation, intelligence, and interest levels in learning, thereby fostering a proactive learning experience for the user. The accuracy of the K-means model decreases as the dataset size increases. When the number of data points reaches 5000, the accuracy drops to 54.7%, which is insufficient for generating personalized learning decisions for students. The SVM teaching system exhibits higher classification accuracy compared to the K-means teaching model, but it still falls short of meeting the standard for autonomous learning.

*5.2. Analysis of AI Instructional Coupling*

In the context of the knowledge economy era, learners should not only be able to obtain useful information from the massive amount of information, but also learn to convert the information into knowledge, link the new knowledge with the existing knowledge and experience, and actively construct new knowledge on the basis of the existing knowledge structure. In this paper, two classes of international law majors are randomly set up as an experimental class and a control class, both of which study in a blended learning environment, with the difference that the experimental class is taught using the AI technology education model, while the control class is not. The two classes were 150 students each. They were assessed based on the study behavior, the completion of homework, and the satisfaction of three aspects of the course to test.

5.2.1. Analysis of Learning Behavior

The data recording function of the teaching cloud platform records in detail students' access to the platform by logging in and watching videos, submitting assignments, and so on. In this paper, the data generated on the platform for a period of eighteen weeks is selected as the object of analysis, and the data of the experimental class and the control class are compared. Table 2 shows the access data of the teaching platform, and the total number of visits of the experimental class is significantly higher than that of the control class, with 7859 visits for the experimental class compared to 6489 visits for the control class. This indicates that students in the experimental class used the teaching platform more frequently and showed a stronger willingness to participate in learning. The average number of visits for the experimental class was 252 compared to 105 for the control class. On average, students in the experimental class accessed the teaching platform more times per student, indicating that students were more actively engaged in learning activities and benefited from the personalized learning support of the AI-coupled empowerment model. The number of comments, views, and likes in the experimental class is significantly higher than that in the control class. This indicates that students in the experimental class are more actively engaged in online interactions, demonstrating a higher level of social interaction. The control class had a relatively lower number of views and likes, indicating that students in the control class were relatively conservative in their interactions. Taken together, the overall engagement of students in the experimental class is significantly higher than that of the control class, which is a positive effect of the innovative exploration of the coupling and empowerment model of AI and international legal education. Students are more willing to obtain learning resources through the teaching platform, which is attributed to the personalized recommendation and support of AI technology, which makes learning closer to students' needs and improves the effect and enthusiasm of learning.

**Table 2.** Teaching platform access data.

| Classes | Class Size | Data Recording | Total Number of Times | Average Times |
|---|---|---|---|---|
| Experimental class | 150 | Number of logins | 7859 | 252 |
| | | Number of visits | 6008 | 128 |
| | | Number of comments | 5078 | 108 |
| | | Number of views | 7950 | 224 |
| | | Number of likes | 6948 | 145 |
| Control class | 150 | Number of logins | 6489 | 105 |
| | | Number of visits | 5190 | 95 |
| | | Number of comments | 4467 | 79 |
| | | Number of views | 2578 | 65 |
| | | Number of likes | 5042 | 82 |

In the teaching of the experimental class, the teacher will appropriately guide the students to watch the teaching video through the AI international legal education coupling empowerment mode in the face-to-face classroom or after-school time, which makes the students watch the video more actively for in-depth learning. Table 3 shows the students' viewing of videos on the AI teaching platform; the longest viewing time is about the same, but the experimental class watched the video for a maximum of 191.9 h, and the maximum time spent on watching the text reached 124.5 h. It can be seen that the teaching strategy implemented in the experimental class can improve the students' initiative in video learning. It shows that AI and international legal education coupling empowerment mode will inevitably increase the speed and quality of talent supply, and the application of the AI coupling empowerment teaching method greatly enriches the means of legal education, stimulates the students' enthusiasm for learning to a certain extent, strengthens the students' practical ability, and improves the quality of legal education.

**Table 3.** Students watching the AI platform video.

| Classes | Data Recording | Maximum Time (h) | Average Time (h) |
|---|---|---|---|
| Experimental class | Watch the video | 191.9 | 189.5 |
| | Watch the text | 124.5 | 122.5 |
| | Length of study | 144.7 | 140 |
| | Length of time logged on | 189.9 | 175.9 |
| Control class | Watch the video | 189.5 | 67.6 |
| | Watch the text | 109 | 77 |
| | Length of study | 126.8 | 98 |
| | Length of time logged on | 105.5 | 68 |

### 5.2.2. Completion of Assignments

Since three teaching contents were selected as experimental teaching cases in this study, the assignments corresponding to the three contents were also analyzed. Teachers strictly set the time for homework submission every time they assigned homework in the experimental class and provided students with eight opportunities to submit their homework. Figure 7 shows the usual homework submission situation, and the experimental class appeared to have a 100% homework submission rate on three occasions, while the control class had a maximum homework submission rate of 72%, and there were cases in which individual students did not submit their homework. It can be seen that the teaching strategy implemented for the experimental class can enable students to actively engage in the production of their works. To realize the coupled and empowered development mode of legal education and AI, it is necessary to break the backward concept, closed mode, single method, and solidified content obstacles existing in legal education at this stage, and promote the development of legal education.

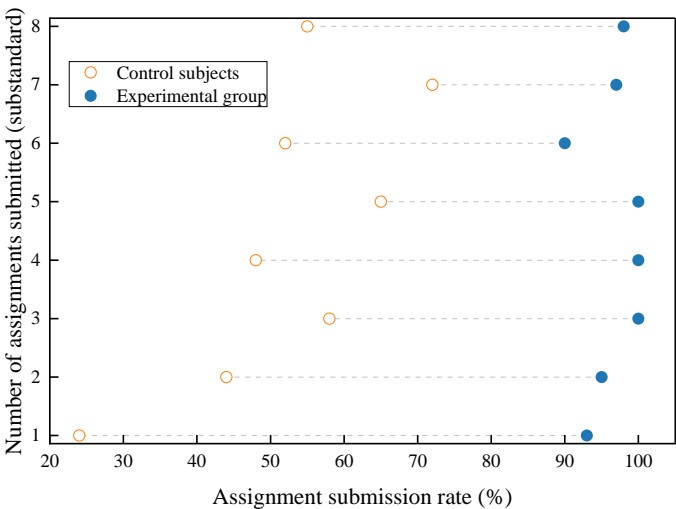

**Figure 7.** The daily assignment submission status.

### 5.2.3. Course Satisfaction

After the end of the semester course, the teacher arranged for the students to fill out the self-assessment scale of the course satisfaction and compared the data of the experimental class group with the data of the control class group, i.e., after studying the course, the independent samples *t*-test was conducted for the students' data of the course satisfaction, and the results of the test are shown in Table 4. The significance of the variance of the two classes is 0.010 < 0.05, so there is a significant difference between the variance of the two classes, and the data in the table shows that the probability of the two-test *t*-test is 0.000 < 0.05 and <0.01, which means that there is an extremely significant difference between the experimental class and the control class in terms of the feelings of course experience. It shows that the experimental class is better than the control class in terms of satisfaction, engagement, and satisfaction with the course. It indicates that full cooperation between law teachers and legal AI can help teachers reduce mechanical repetitive labor, such as the student data set recording and analysis of test assignments in the basic teaching session, which verifies that AI coupling empowerment plays an important role in the transformation and upgrading of legal education.

**Table 4.** Results of the course satisfaction test.

| Classes | N | Data Validation | Average | Standard Deviation | Standard Error of the Mean |
|---|---|---|---|---|---|
| Experimental class | 150 |  | 53.47 | 6.248 | 0.911 |
|  |  | T | 0.008 | 0.004 | 0.001 |
|  |  | P | 0.007 | 0.002 | 0.001 |
| Control class | 150 |  | 47.38 | 5.648 | 0.701 |
|  |  | T | 0.025 | 0.018 | 0.05 |
|  |  | P | 0.048 | 0.016 | 0.05 |

### 5.3. Long-Term Impact and Sustainability Analysis

#### 5.3.1. Follow-Up Studies

By tracking the academic and career development of the students in the experimental group over time, we understand the long-term impact of the AI-coupled empowerment model on the students' future. Table 5 shows the tracking statistics of the graduates' data. The students in the experimental group have maintained a high level of academic development, with their GPA gradually increasing from 3.8 to 5.0, while the academic development of the control group is relatively slower, growing from 3.2 to 4.5. The continuous improvement of the experimental group's students in academic development reflects the positive

impact of the AI-coupled empowerment model on the transfer of knowledge and in-depth learning. The career development of students in the experimental group showed a clear advantage in terms of employment rate, which rapidly increased from 75% to 100%, while the control group grew from 65% to 95%. It shows that the AI-coupled empowerment model has a positive effect on improving students' career competitiveness and adaptability in the field of law, making it easier for them to find jobs that match their professional background. The experimental group showed a favorable positive trend in academic and career development, while the control group grew relatively slowly in comparison. The long-term trend suggests that the innovative exploration of international legal education by the AI-coupled empowerment model has achieved significant results in developing students' academic depth and career application skills. Despite the gap between the control group relative to the experimental group, students in the control group also made significant progress over the 10 graduation years. Reflecting the overall improvement in the field of international law, it also emphasizes the leading edge of the experimental group in academic and career development.

**Table 5.** Graduate data tracking.

| Years of Graduation | Experimental Class | | Control Class | |
|---|---|---|---|---|
| | Academic Development | Career Development (Employment Rate) | Academic Development | Career Development (Employment Rate) |
| 1 | 3.8 | 75% | 3.2 | 65% |
| 2 | 3.9 | 80% | 3.4 | 70% |
| 3 | 4.1 | 85% | 3.5 | 75% |
| 4 | 4.2 | 90% | 3.6 | 80% |
| 5 | 4.5 | 95% | 3.8 | 81% |
| 6 | 4.8 | 98% | 4.0 | 85% |
| 7 | 5.0 | 100% | 4.1 | 90% |
| 8 | 5.0 | 100% | 4.2 | 92% |
| 9 | 5.0 | 100% | 4.4 | 94% |
| 10 | 5.0 | 100% | 4.5 | 95% |

5.3.2. Sustainability Assessment

Statistics on the number of times and success rate of students' participation in actual jurisprudence practice cases are used to assess the impact of AI on practical application ability. Figure 8 shows the students' participation in law time, and the number of times students in the experimental group participated in actual law practice cases increased year by year, growing from the initial 3 times to 12 times, indicating that the AI-coupled empowerment model played a positive role in stimulating students' active participation in practice. The success rate of the students in the experimental group increased year by year, from 80% to 100%, while that of the students in the control group was relatively slow, from 70% to 96%, and the success rate of the students in the experimental group in practical applications was significantly better than that of the control group. The success rate of students in the experimental group exceeded that of the control group from the second year onwards, emphasizing the positive impact of the AI-coupled empowerment model on the students' ability in practical applications. Students in the experimental group achieved a 95% success rate in year 4, while the control group reached a similar level only in year 7, indicating that the AI coupling empowerment model accelerated the development of students' early practical application ability. In the long run, the students in the experimental group maintained a high level of both the number of engagements and the success rate, reflecting the sustainability advantages of the AI-coupled empowerment model for real-world application skills in the long run.

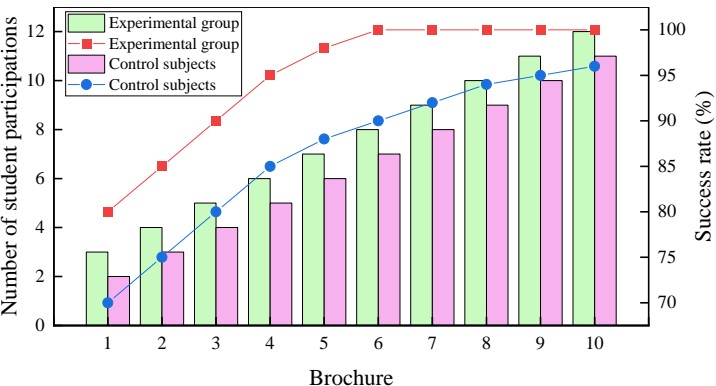

**Figure 8.** Participation in the practice of law.

Here, we present statistics on student feedback on actual law practice, including ratings of the ease and usefulness of applying actual cases. Students gave subjective ratings of the difficulty of actual law practice, with 1 indicating very difficult and 10 indicating very easy. Students also gave subjective ratings of the practicality of actual law practice, with 1 indicating very impractical and 10 indicating very practical. Table 6 shows the results of the evaluation of the degree of difficulty and practicality. The experimental group students' ratings of the degree of difficulty of practical law practice increased year by year, gradually increasing from 7 to 10. It shows that students in the experimental group made gradual progress in the more difficult cases, showing the effectiveness of the AI-coupled empowerment model in guiding students to overcome challenges. Students in the experimental group consistently maintained a score of 10 on the utility scale, indicating that students perceived the practices they learned to be very highly useful for their future law careers, helping them to better cope with career challenges and demands. Highlighting the positive impact of the sustainability of the AI-coupled empowerment model in actual law practice, students in the experimental group consistently demonstrated superiority in terms of difficulty and utility.

**Table 6.** Evaluation of difficulty and practicability.

| Number of Practices | Experimental Class | | Control Class | |
|---|---|---|---|---|
| | Degree of Difficulty | Practicality | Degree of Difficulty | Practicality |
| 1 | 7 | 8 | 6 | 7 |
| 2 | 7.5 | 8.5 | 6.5 | 7.5 |
| 3 | 8 | 9 | 7 | 8 |
| 4 | 8.5 | 9.5 | 7.5 | 8.5 |
| 5 | 9 | 10 | 8 | 9 |
| 6 | 9.5 | 10 | 8.5 | 9.5 |
| 7 | 9.8 | 10 | 9 | 9.8 |
| 8 | 10 | 10 | 9.2 | 9.9 |
| 9 | 10 | 10 | 9.5 | 9.9 |
| 10 | 10 | 10 | 9.8 | 10 |

## 6. Discussion

In practice, despite the potential empowering role of AI technology in international legal education, its application faces certain limitations. Firstly, the dissemination and utilization of AI tools rely on the level of digitization within educational institutions and among students, giving rise to issues related to the digital divide. Secondly, the decision-making process of AI is often opaque, making it challenging to explain the specific basis for its recommendations or decisions. This may raise concerns in legal education regarding transparency and interpretability. Additionally, considering the complexity of the field of international legal studies, AI models may struggle to comprehensively cover all dimensions of legal knowledge, resulting in a partial and incomplete understanding.

## 7. Conclusions

In this paper, AI technology is combined with international law education to construct an AI-coupled empowerment education model. The effectiveness of the model is verified in practical teaching, and the conclusions are as follows.

1.  In the teaching system test, the accuracy of the classification of the AI international law education model reaches 99.5%, and at the same time, the algorithm runs in a relatively short time, which is able to mine the learning knowledge for the user more accurately in a short period of time.
2.  In the coupling analysis, the average number of visits in the experimental class is 128, and the final score of the student who has watched the video for the longest time is also 13 points more than that of the control class, and there is a situation in which the submission rate of three assignments is 100%. The coupling of legal education and AI to empower development can help teachers to reduce mechanical repetitive labor.
3.  In the long-term impact and sustainability analysis, the career development of students in the experimental group showed a clear advantage in terms of the employment rate, which rapidly increased from 75% to 100%, and they achieved a 95% success rate in the fourth year. The positive impact of the sustainability of the AI-coupled empowerment model in actual law practice is highlighted.

**Author Contributions:** Conceptualization, S.Y.; Methodology, S.Y.; Validation, Y.W.; Formal analysis, Y.W.; Investigation, Y.W.; Resources, S.Y.; Data curation, Y.W.; Writing—original draft, Y.W.; Writing—review & editing, S.Y.; Supervision, S.Y.; Funding acquisition, S.Y. All authors have read and agreed to the published version of the manuscript.

**Funding:** This research received no external funding.

**Institutional Review Board Statement:** Not applicable.

**Informed Consent Statement:** Informed consent was obtained from all subjects involved in the study.

**Data Availability Statement:** The data presented in this study are available on request from the corresponding author.

**Conflicts of Interest:** The authors declare no conflict of interest.

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
