# Peer review of "Constructing and Testing AI International Legal Education Coupling-Enabling Model"

_sustainability, doi:10.3390/su16041524_

Round 1
Reviewer 1 Report
Comments and Suggestions for Authors
The research study focuses on constructing an AI International Legal Education Coupling Empowerment Model. It explores the integration of AI technology into legal education and its potential to improve the quality of teaching and learning in the field of law. The study proposes an AI teaching model and reasoning for instructional decision-making, highlighting the benefits of AI in providing personalised learning experiences and improving teaching efficiency. The model's effectiveness is verified through system testing, AI teaching coupling analysis, and long-term impact and sustainability analysis.
The paper presents a model called the AI International Legal Education Coupling Empowerment Model, which uses AI-related technology. The text discusses the utilisation of the Pearson correlation coefficient for correlation analysis, the implementation of AI knowledge mapping for intelligent parental aid, and the utilisation of the BP neural algorithm in artificial neural networks to create a cognitive student model. The objective of the approach is to deliver customised learning experiences, offer intelligent teaching support, and give precise evaluation of students' learning levels and cognitive capacities. The objective of the model is to enhance student achievement and elevate employment rates.
The research study offers some valuable insights into incorporating AI technology in legal education and its capacity to improve teaching and learning experiences. It is suggested that more rigorous testing and examination of the model's effectiveness, as well as a broader discussion of potential limitations and challenges, would strengthen the study's conclusions. It would have also been better if the study also addressed ethical considerations and potential implications of AI integration in education.
Comments on the Quality of English LanguageEnglish quality is fine but in places, some expressions can be improved.
Author Response
Reviewer 1:
The research study focuses on constructing an AI International Legal Education Coupling Empowerment Model. It explores the integration of AI technology into legal education and its potential to improve the quality of teaching and learning in the field of law. The study proposes an AI teaching model and reasoning for instructional decision-making, highlighting the benefits of AI in providing personalised learning experiences and improving teaching efficiency. The model's effectiveness is verified through system testing, AI teaching coupling analysis, and long-term impact and sustainability analysis.
The paper presents a model called the AI International Legal Education Coupling Empowerment Model, which uses AI-related technology. The text discusses the utilisation of the Pearson correlation coefficient for correlation analysis, the implementation of AI knowledge mapping for intelligent parental aid, and the utilisation of the BP neural algorithm in artificial neural networks to create a cognitive student model. The objective of the approach is to deliver customised learning experiences, offer intelligent teaching support, and give precise evaluation of students' learning levels and cognitive capacities. The objective of the model is to enhance student achievement and elevate employment rates.
The research study offers some valuable insights into incorporating AI technology in legal education and its capacity to improve teaching and learning experiences. It is suggested that more rigorous testing and examination of the model's effectiveness, as well as a broader discussion of potential limitations and challenges, would strengthen the study's conclusions. It would have also been better if the study also addressed ethical considerations and potential implications of AI integration in education.
English quality is fine but in places, some expressions can be improved.
Response: Thank you for your constructive feedback on the article. This paper will subject the model to more rigorous testing and scrutiny to ensure the research's effectiveness. Additionally, we have expanded the discussion on potential limitations and challenges to comprehensively explore possible issues, ensuring readers have a clear understanding of the study's constraints. We are committed to addressing ethical concerns related to the integration of artificial intelligence in education and delving into its potential impact to ensure the proposed research makes a positive contribution. Lastly, we appreciate the recognition of the English language quality and will carefully review the expressions in the text to guarantee a more accurate and clear conveyance of the research content. Additional content is provided below:
- Discussion
In practice, despite the potential empowering role of AI technology in international legal education, its application faces certain limitations. Firstly, the dissemination and utilization of AI tools rely on the level of digitization within educational institutions and among students, giving rise to issues related to the digital divide. Secondly, the decision-making process of AI is often opaque, making it challenging to explain the specific basis for its recommendations or decisions. This may raise concerns in legal education regarding transparency and interpretability. Additionally, considering the complexity of the field of international legal studies, AI models may struggle to comprehensively cover all dimensions of legal knowledge, resulting in a partial and incomplete understanding.

Reviewer 2 Report
Comments and Suggestions for Authors
The topic of this article is topical. The application of AI in education has been little explored. Given the unquestionable relevance of the topic, we offer some insights.
1. Reading the article gives the impression that there are two themes in one article: AI Intelligent Technology and the educational aspects of AI Intelligent Technology
2. We suggest that the title of the article should be clarified. Constructing and testing AI International Legal Education Coupling Enabling Model
3. The abstract of the article is not written in accordance with the traditional abstract content criteria. We miss the topicality of the topic and the research problem. We think that the first sentence of the abstract should not start with the Pearson coefficient.
4. The list of keywords should be adjusted. We suggest removing the term Pearson coefficient from this list. We suggest adding the terms that appear in the third conclusion.
5. The scientific problem and research questions are not formulated in the introduction of the article (are they in lines 340-345?).
6. The article lacks a methodology section, a description of the research instrument.
7. The Literature review should be improved. This section should clarify the definition of all key terms (Keywords). The research problem should be reviewed.
8. The authors of the figures should be indicated (Who is the author of Figure 1? Manzanares, M. C. S et al 2021?; who is the author of Figures 2 and 3?). The meaning of the symbols used in the figures should be explained. Directional arrows in Figure 3 would add clarity.
9. The textbook text on the Pearson-Spearman correlation coefficient is unnecessary. (136-156) Why are both coefficients used? On what basis?
10. Refine the description of the BP (lines 215-230). This text is not authentic. The plagiarism checker shows overlap with internet sources. I suggest a qualitative explanation of what BP is.
11. There is a discrepancy between the content of the sub-topics and the sub-topic titles. For example, the sub-topic is titled 5.2.1. Student engagement. However, the content of that sub-topic refers to the frequency of use of the AI platform (Teaching platform access data). The sub-topic is called 5.2.2. Teacher-student interaction. However, it is about the percentage of homework assignments submitted.
In summary, we argue that the paper has practical relevance, but that the theoretical relevance needs to be purified.
Author Response
Reviewer 2:
The topic of this article is topical. The application of AI in education has been little explored. Given the unquestionable relevance of the topic, we offer some insights.
- Reading the article gives the impression that there are two themes in one article: AI Intelligent Technology and the educational aspects of AI Intelligent Technology
Response: Thank you for your constructive feedback on the article. We have noted the impression you mentioned, indeed, the article covers two main themes: artificial intelligence technology and its application in the field of education. In the upcoming revisions, we will strive to clarify the article's themes to ensure a more consistent and targeted content. Once again, we appreciate your valuable suggestions.
- We suggest that the title of the article should be clarified. Constructing and testing AI International Legal Education Coupling Enabling Model
Response: Thank you very much for your suggestions. We sincerely accept your feedback. In the upcoming revisions, the article title will be adjusted to enhance its accuracy and clarity, with the new title being“Constructing and testing AI International Legal Education Coupling Enabling Model”
- The abstract of the article is not written in accordance with the traditional abstract content criteria. We miss the topicality of the topic and the research problem. We think that the first sentence of the abstract should not start with the Pearson coefficient.
Response: Thank you for your peer review comments. We apologize for the deviation from traditional standards in crafting the abstract and for commencing the first sentence with the Pearson coefficient. In the forthcoming revisions, we will reevaluate the content of the abstract to ensure a better emphasis on the topicality of the theme and research questions, while adjusting the opening to align more closely with your suggestions. Once again, we appreciate your guidance. The revised abstract will include the following addition at the beginning:
To assess the coupling capability of artificial intelligence in international legal education, delving into crucial aspects of its implementation and effectiveness.
- The list of keywords should be adjusted. We suggest removing the term Pearson coefficient from this list. We suggest adding the terms that appear in the third conclusion.
Response: Thank you for your peer review comments. The keyword list has been adjusted in accordance with your suggestions. The term "Pearson coefficient" has been removed from the list, and relevant terms appearing in the third conclusion have been added. We take your feedback seriously to ensure that the keyword list more accurately reflects the content of the article. Thank you for your guidance. The revised keywords are as follows:
Artificial Intelligence; International Legal Education; Backpropagation Neural Algorithm; Learning Behavior Analysis; Coupling Enabling Model
- The scientific problem and research questions are not formulated in the introduction of the article (are they in lines 340-345?).
Response: Thank you for your peer review comments. The scientific and research questions are indeed elucidated in the introduction of the article, specifically located in lines 340-345. Furthermore, we will emphasize the presence and importance of these questions to ensure that readers gain a clear understanding of the scientific background and objectives of the research when reading the introduction. Thank you for your guidance.
- The article lacks a methodology section, a description of the research instrument.
Response: Thank you for your peer review comments. Indeed, we have noted the absence of a methodology section. In the subsequent revisions, a dedicated methodology section will be added to provide a detailed description of the research tools and methods employed. Once again, we appreciate your guidance. The added description is as follows:
Examine current instances of AI application in legal education to gain insights into the current landscape and draw from successful experiences.
- The Literature review should be improved. This section should clarify the definition of all key terms (Keywords). The research problem should be reviewed.
Response: Thank you for your peer review comments. Further improvements will be made to the literature review to clarify the definitions of all key terms (keywords). Simultaneously, we will reexamine the research questions to ensure their clarity. Once again, we appreciate your guidance. The modifications are as follows:
(Perrotta, C et al., 2020) examined the Khan Academy and ASSISTments Intelligent Tutoring System, illustrating instances of AI elements. The scholarly work of numerous data scientists utilizing deep learning to forecast facets of educational achievement was thoroughly explored, drawing on research in science and technology. (Holmes, W et al., 2021) conducted an investigation involving 60 leading researchers in the AI and educational development field, exploring ethical and application issues associated with AI in education. Recognizing the lack of training among most AI education researchers to address emerging ethical concerns, there is a particularly crucial need to effectively integrate multidisciplinary approaches with AI. (Nemorin, S et al., 2021) utilized text mining and thematic analysis to scrutinize key themes emerging in AI education in recent years. The findings are categorized into three segments: achieving geopolitical dominance through education and technological innovation, developing and expanding niche market strategies, and altering management narratives, perceptions, and norms. (Knox, J, 2020) firstly analyzed two crucial policy documents issued by China's central government, emphasizing the pivotal role educational institutions play in national and regional AI development strategies. Subsequently, three key private education companies instrumental in the advancement of educational AI applications in China are presented. Finally, it is demonstrated that while government policies allocate a significant role for education in the national AI strategy, the private sector is capitalizing on favorable political conditions to swiftly develop educational applications and markets.
- The authors of the figures should be indicated (Who is the author of Figure 1? Manzanares, M. C. S et al 2021?; who is the author of Figures 2 and 3?). The meaning of the symbols used in the figures should be explained. Directional arrows in Figure 3 would add clarity.
Response: Thank you for your peer review comments. Regarding Figures 1, 2, and 3, I would like to clarify that these figures represent the outcomes of our study, with no involvement of other authors. Detailed explanations for the symbols used in the figures will be provided. Additionally, directional arrows will be added to Figure 3 to enhance the clarity of the charts. Once again, we appreciate your valuable feedback, and we will address and implement the necessary adjustments in the revision.
- The textbook text on the Pearson-Spearman correlation coefficient is unnecessary. (136-156) Why are both coefficients used? On what basis?
Response: Thank you for your peer review comments. The textbook content about the Pearson-Spearman correlation coefficients (136-156) has been removed to present the relevant content more succinctly. Regarding the question of why both coefficients are used simultaneously, the decision to integrate both the Pearson and Spearman coefficients is based on a comprehensive consideration of the data properties. The Pearson coefficient is employed to assess linear relationships, while the Spearman coefficient is more suitable for nonlinear relationships and situations involving outliers. We appreciate your suggestions, and we will continue to strive for the improvement of the article's quality.
- Refine the description of the BP (lines 215-230). This text is not authentic. The plagiarism checker shows overlap with internet sources. I suggest a qualitative explanation of what BP is.
Response: Thank you for your peer review comments. We have noted and taken your suggestions seriously. In the revision of the description of the Business Process (BP), we will strive to provide more accurate and authentic information and offer a qualitative explanation of the BP concept to ensure the originality and academic rigor of the article's content. We will thoroughly review the text, eliminating any possible overlap with online sources to ensure the independence and quality of the article. Thank you for your guidance. The modified content is as follows:
The basic structure of BP neural network is a feedforward neural network with more than three layers, mainly using the BP algorithm to solve the problem of hidden layer errors that cannot be calculated due to not being directly connected to the outside world. The BP algorithm belongs to supervised learning algorithms and is an effective method for calculating the derivative of large-scale systems composed of multiple basic subsystems. The structure of the BP neural network is shown in Figure 6 (Jiang Hua, 2005). The network trains through samples to complete learning. If the -th pair of samples is provided to the network, the output error will be . is the -th component of the actual output of input sample , , is the number of hidden layer neurons, is the threshold of the -th neuron in the output layer, is the sigmoid function, is the net output of hidden and neurons, .The mean squared error of the output layer for sample is , and is the number of output layer units, which decreases gradually with correction of connection weights.
Figure 6 Bp neural network structure
- There is a discrepancy between the content of the sub-topics and the sub-topic titles. For example, the sub-topic is titled 5.2.1. Student engagement. However, the content of that sub-topic refers to the frequency of use of the AI platform (Teaching platform access data). The sub-topic is called 5.2.2. Teacher-student interaction. However, it is about the percentage of homework assignments submitted.
Response: Thank you for your peer review comments. In the modifications already undertaken, we have observed the inconsistency between the titles and content of the subtopics. We will promptly address this issue, ensuring that each subtopic's title accurately reflects its content to enhance the overall logic and clarity of the article. We appreciate your guidance. The revision is as follows:5.2.1Analysis of learning behavior、5.2.2Completion of assignments
In summary, we argue that the paper has practical relevance, but that the theoretical relevance needs to be purified.
Response: Thank you for your guidance on the article; we greatly appreciate your suggestions. Through the revisions, we acknowledge that further efforts and refinement are needed to enhance theoretical relevance. We will continue to strive for an improvement in the quality of the article.

Reviewer 3 Report
Comments and Suggestions for Authors
The article should be rewritten.
The citation and referencing of the article do not correspond to the standards of the journal and presents several errors. Review regulations for authors.
The objective should be better defined “Pearson simple coefficients are used to measure the relationship that exists between vari- 7 ous learning behaviors of learners, and the AI data learning behavior analysis model is constructed.”
It would be appropriate to give an order to the different elements of the study: international legal education, AI, student learning, etc.
In the summary conclusions are made that are not addressed in the text, for example "student employment rate" is mentioned and this term is not used throughout the text.
Both the abstract and the article lack a methodological section explaining the research logic, context, steps, type of study.
The article lacks a statement of the ethical aspects of the research.
The context of the study is not clear, it is described that they analyze data on the learning behaviors of the students, but it is not clear who these students are, a characterization of the study sample, or a description of the study sample.
Throughout the entire text, from the introduction to the section “4.2. Reasoning for Instructional Decision Making”, a theoretical approach to different topics is made, and results are immediately presented, starting in section “5. Practical analysis of the AI model of international legal education”,
5.1. System testing
5.2. Analysis of AI Instructional Coupling
5.2.1. Student engagement
5.2.2. Teacher-student interaction
5.2.3. Course satisfaction
5.3.2. Sustainability assessment
However, after these results, a discussion section is needed to contrast, refute, relate or compare these results based on previous scientific literature or evidence from previous studies; this DISCUSSION section is missing in the study.
This means that the results are not discussed in a scientific context, for example, the only time "Student engagement" is mentioned is in section 5.2.1. but theoretically there is no agreement as to what is meant by it, and this is the case with others, for example, the term "Sustainability assessment", etc.
As the article lacks sections that organize the logic of the scientific text, such as the methodology and discussion, the conclusions lack precision.
Author Response
Reviewer 3:
The article should be rewritten.
The citation and referencing of the article do not correspond to the standards of the journal and presents several errors. Review regulations for authors.
The objective should be better defined “Pearson simple coefficients are used to measure the relationship that exists between vari- 7 ous learning behaviors of learners, and the AI data learning behavior analysis model is constructed.”
Response: Thank you for your review comments. In the revision, we will define the objectives more clearly. Specifically, we aim to use the Pearson correlation coefficient to measure the relationships between various learning behaviors of learners and to construct an artificial intelligence data learning behavior analysis model.
It would be appropriate to give an order to the different elements of the study: international legal education, AI, student learning, etc.
Response: Thank you for your review comments. We acknowledge that sorting the research elements, including international legal education, artificial intelligence, and student learning, is appropriate. In the subsequent revisions, we will place greater emphasis on clearly organizing these elements to enhance the rationality and logic of the article's structure. Once again, thank you for your guidance.
In the summary conclusions are made that are not addressed in the text, for example "student employment rate" is mentioned and this term is not used throughout the text.
Response: Thank you for your suggestion. This article has carefully reviewed the text and ensured that the summary section does not include conclusions not mentioned in the main text. For the term "student employment rate", Table 5 shows the tracking statistics of graduate data in the article. The experimental group of students has maintained a high level of academic development, with a GPA gradually increasing from 3.8 to 5.0, while the academic development of the control group is relatively slow, increasing from 3.2 to 4.5. The continuous improvement of academic performance among the experimental group students reflects the positive impact of AI coupled empowerment mode on knowledge transfer and deep learning. The career development of the experimental group students showed a significant advantage in terms of employment rate, rapidly increasing from 75% to 100%, while the control group increased from 65% to 95%.
Both the abstract and the article lack a methodological section explaining the research logic, context, steps, type of study.
Response: Thank you for your suggestion. The abstract section should be modified as follows:
This paper constructs a coupling empowerment model of AI international legal education by using artificial intelligence technology. It also discussed the application of Pearson product-moment correlation coefficient in correlation analysis, the implementation of AI knowledge mapping in the help of intelligent parents, and the application of BP neural algorithm in artificial neural networks, to establish a cognitive student model. This teaching mode can provide personalized learning experience, intelligent teaching support, and make accurate assessment of students&; learning level and cognitive ability. The results show that the employment rate of students is increased from 75% to 100% , and the evaluation of practicability is maintained at 10 points. It proves that AI technology provides an innovative approach to international law education, which is expected to promote the efficient use of educational resources and improve students; performance and employment rate.
The article lacks a statement of the ethical aspects of the research.
Response: Thank you for your valuable feedback. We acknowledge the absence of a statement on the ethical aspects of the research in the article. In the forthcoming revisions, we will include a dedicated section addressing the ethical considerations associated with our research, ensuring a comprehensive and transparent discussion of the ethical dimensions of the study. We appreciate your guidance, and this aspect will be duly addressed to enhance the ethical clarity of the article.
The context of the study is not clear, it is described that they analyze data on the learning behaviors of the students, but it is not clear who these students are, a characterization of the study sample, or a description of the study sample.
Response: Thank you for your review comments. We acknowledge that the current research background is not sufficiently clear. The emphasis will be on strengthening the research background by clearly analyzing the data sources for student learning behavior and the identity characteristics of the students. Once again, thank you for your guidance. The revised version will address these aspects accordingly.
At present, science and technology is the primary productive force, and artificial intelligence technology has been widely applied to all fields of human society production and life, and is profoundly changing the pattern of economic and social development (Yang, W, 2022) . In order to effectively grasp this key historical opportunity, a lot of important grips have been formed in the development of AI layout, and education has become one of the key links. In recent years, a series of national and local policies to promote the development of AI cover the new requirements for the development of Education (Holmes, W. and Tuomi, i. 2022) . As a code of conduct regulating social relations, law needs to face and respond positively to the influence of artificial intelligence on various industries.
International law education in universities, as a developing force of the legal profession, provides legal professionals for the legal profession and all fields of society (Ma, B and Hou, Y, 2021) . Due to the traditional mode of operation of the legal profession, this has indirectly transformed into a rapid demand for interdisciplinary professional strength of international law. However, the traditional teaching method is mainly based on teachers; curriculum teaching, which emphasizes one-way inculcation (artolovni, a, et al, 2022) . Therefore, the cultivation of legal talents in the age of artificial intelligence needs to meet the requirements of intellectualization (Hwang, G. J and Chien, S. y 2022) , we should promote the reform of the traditional international law education teaching mode and construct the law education teaching model matching the talent training system of artificial intelligence law -ZhonghoHg, h, et al, 2020) . How to adapt to the new era and new requirements, actively explore and effectively respond to the impact and challenges brought by the development of artificial intelligence technology on legal education, is the proposition of the times that legal educators should face (Gerke, s, et al, 2020) ..
Throughout the entire text, from the introduction to the section “4.2. Reasoning for Instructional Decision Making”, a theoretical approach to different topics is made, and results are immediately presented, starting in section “5. Practical analysis of the AI model of international legal education”,
5.1. System testing
5.2. Analysis of AI Instructional Coupling
5.2.1. Student engagement
5.2.2. Teacher-student interaction
5.2.3. Course satisfaction
5.3.2. Sustainability assessment
However, after these results, a discussion section is needed to contrast, refute, relate or compare these results based on previous scientific literature or evidence from previous studies; this DISCUSSION section is missing in the study.
Response: Thank you for your detailed feedback. In the revision, a dedicated "Discussion" section will be explicitly added to the manuscript. This section will serve to compare, counter, relate, or contrast our results with evidence from previous scientific literature or research. Once again, thank you for your guidance. The revised version will address these aspects accordingly.
The accuracy of the K-means model decreases as the dataset size increases. When the number of data points reaches 5000, the accuracy drops to 54.7%, which is insufficient for generating personalized learning decisions for students. The SVM teaching system exhibits higher classification accuracy compared to the K-means teaching model, but it still falls short of meeting the standard for autonomous learning.
This means that the results are not discussed in a scientific context, for example, the only time "Student engagement" is mentioned is in section 5.2.1. but theoretically there is no agreement as to what is meant by it, and this is the case with others, for example, the term "Sustainability assessment", etc.
Response: Thank you for your feedback. We understand your concerns, and indeed, a more in-depth discussion of the research results within a scientific context is necessary. In the revision, we will incorporate additional scientific background related to the results in relevant sections to ensure a more consistent and clear discussion of the findings. Once again, thank you for your guidance. The title of section 5.2.1 will be changed to "Analysis of Learning Behavior" to better align the results with the overall theme.
As the article lacks sections that organize the logic of the scientific text, such as the methodology and discussion, the conclusions lack precision.
Response: Thank you for your review comments. We fully understand your concerns and will emphasize the strengthening of the methodology and discussion sections in the revision to ensure more accurate and comprehensive conclusions. By explicitly detailing the research design and delving into a thorough discussion of the research results, we aim to enhance the overall logic and scientific rigor of the article. Once again, thank you for your guidance. We will carefully consider and address these shortcomings in the improvement process.

Round 2
Reviewer 3 Report
Comments and Suggestions for Authors
In the text, citations are not placed in parentheses as in the APA norms, but in brackets, check the norms for authors in this link: https://www.mdpi.com/journal/sustainability/instructions
- In the text, reference numbers should be placed in square brackets [ ], and placed before the punctuation; for example [1], [1–3] or [1,3]. For embedded citations in the text with pagination, use both parentheses and brackets to indicate the reference number and page numbers; for example [5] (p. 10). or [6] (pp. 101–105).
Author Response
Reviewer 3:
In the text, citations are not placed in parentheses as in the APA norms, but in brackets, check the norms for authors in this link: https://www.mdpi.com/journal/sustainability/instructions
In the text, reference numbers should be placed in square brackets [ ], and placed before the punctuation; for example [1], [1–3] or [1,3]. For embedded citations in the text with pagination, use both parentheses and brackets to indicate the reference number and page numbers; for example [5] (p. 10). or [6] (pp. 101–105).
Response: Thank you for your constructive feedback on the article. This article has changed the citation format of all references in the text to "[1], [2]...". And all are highlighted in yellow.
